# Crescents and IgA Nephropathy: A Delicate Marriage

**DOI:** 10.3390/jcm11133569

**Published:** 2022-06-21

**Authors:** Hernán Trimarchi, Mark Haas, Rosanna Coppo

**Affiliations:** 1Nephrology Service, Hospital Britanico de Buenos Aires, Buenos Aires C1280 AEB, Argentina; htrimarchi@hotmail.com; 2Department of Pathology and Laboratory Medicine, Cedars-Sinai Medical Center, Los Angeles, CA 90048, USA; 3Fondazione Ricerca Molinette, Regina Margherita Hospital, 10126 Turin, Italy; rosanna.coppo@unito.it

**Keywords:** IgA nephropathy, crescents, proteinuria, glomerular filtration rate, Oxford score

## Abstract

IgA nephropathy (IgAN) is a progressive disease with great variability in the clinical course. Among the clinical and pathologic features contributing to variable outcomes, the presence of crescents has attracted particular interest as a distinct pathological feature associated with severity. Several uncontrolled observations have led to the general thought that the presence and extent of crescents was a prognostic indicator associated with poor outcomes. However, KDIGO 2021 guidelines concluded that either the presence or the relative number of crescents should not be used to determine the progression of IgAN nor should they suggest the choice of immunosuppression. Our aim is to report and discuss recent data on the debated issue of the value of active (cellular and fibrocellular) crescents in the pathogenesis and clinical progression of IgAN, their predictive value, and the impact of immunosuppression on renal function. We conclude that the value of crescents should not be disregarded, although this feature does not have an independent predictive value for progression in IgAN, particularly when considering immunosuppressed patients. An integrated overall evaluation of crescents with other active MEST scores, clinical data, and novel biomarkers must be considered in achieving a personalized therapeutic approach to IgAN patients.

## 1. Introduction

IgA nephropathy (IgAN), defined by prevalent mesangial IgA deposits [1], displays a variable clinical course, ranging from persistent mild microscopic hematuria with or without mild proteinuria and normal renal function to nephrotic syndrome or, rarely, a rapidly progressive course of kidney function loss. Moreover, in approximately 30% of cases, it leads to end-stage kidney disease over a 20-year follow-up period [2]. This variability represents a challenge for the clinician and has elicited the search for clinical and pathologic features predictive of progression [3] but definitively demonstrates that IgAN cannot be considered as a benign condition.

Shortly after the identification of IgAN as a histological pattern of glomerular injury with a variable clinical course, the presence of crescents attracted particular interest as a distinct pathological feature associated with severity. In small but well-described case series, crescents were found to be present in patients with macroscopic hematuria and worse renal function [4] and with a rapid progressive course with loss of renal function in cases with more than 30% of glomeruli involved with crescents [5]. A relationship between crescents and transplanted kidney dysfunction with loss of grafts in patients with recurrent IgAN was also reported [6] and further supported by the observation of an increased recurrence rate of in patients with crescentic forms of IgAN in their native kidneys, despite baseline immunosuppressive therapy [7]. Moreover, the value of crescents to classify children with IgA vasculitis nephritis (IgAVN) established by the International Study for Kidney Disease in Children (ISKDC) [8] suggested a possible similar role for crescents as a histologic marker and as a potential tool to assess treatment decisions concerning immunosuppression in primary IgAN, due to the strong similarities between both entities.

These observations have led to the general thought that finding crescents in kidney biopsies of IgAN cases was a prognostic indicator associated with poor outcomes [9]. Furthermore, results from many clinical studies supported the value of crescents as markers of progression that benefited from immunosuppressive approaches [10,11].

Interestingly, the predictive value of crescents was difficult to demonstrate with recent sophisticated statistical approaches on large cohorts enrolling patients with or without corticosteroid-immunosuppressive treatments, since patients with crescents were those most frequently treated [12]. Other reasons that complicate this analysis are the different definitions employed to identify crescents; the inclusion of extra-capillary proliferation with either cellular, fibrocellular, or fibrous components; and the relative number of crescents in relation to the total number of glomeruli encountered in a kidney biopsy. Different prognostication models employing artificial intelligence provided differing conclusions regarding the inclusion of crescents as predictive markers of progression in IgAN [13]. In this regard, the variability of these results led the Kidney Disease Improving Global Outcomes (KDIGO) 2021 guidelines for the management of IgAN to recommend that either the presence or the relative number of crescents should not be used to determine the likely progression of IgAN nor should they suggest the choice of immunosuppression [14]. KDIGO also made the controversial proposal that even the presence of crescents in >50% of glomeruli in a kidney biopsy in the absence of a concomitant change in glomerular filtration rate (GFR) does not constitute a rapidly progressive situation and ought not to bias therapy.

Despite these negative authoritative considerations, for most nephrologists, the value of crescents remains an unsolved conundrum, since it is commonly recognized that cellular crescents are often associated with a rapid progression of kidney disease and clinically severe proteinuria and hematuria in patients with IgAN, particularly when the relative number is high in a kidney biopsy. Unfortunately, the issue of the value of crescents in IgAN cannot be explored in experimental IgAN, since in these models mostly mesangial proliferation and glomerular sclerosis, but not crescents, are reproduced [15].

The aim of this review is to report recent data on the debated issue of the value of active (cellular and fibrocellular) crescents in the pathogenesis and clinical progression of IgAN, their predictive value as a histological marker, and the impact of immunosuppressive treatment on renal function.

## 2. Molecular Factors Involved in Crescent Formation in IgAN

In IgAN, fibrinogen and fibrin-related molecules are present in crescents (Figure 1). Moreover, components of the glomerular basement membrane, such as collagen types IV and V, laminin, fibronectin, and cytokeratin, are persistently positive within all stages of crescents. Vimentin, usually located in podocytes and parietal epithelial cells as well as in interstitial cells, is also present in all stages of crescents in IgAN, suggesting that at early stages of crescent development in IgAN podocytes may play a key role. In addition, the presence of intrinsic basement membrane constituents is consistent with GBM rupture, likely related at least in part to necrotizing lesions within the glomerular tuft, representing a precursor to crescent formation. Fibrinoid necrosis may be seen in IgAN, most often in cases with active crescentic lesions, although in most studies from Europe and North America, this is relatively uncommon, especially when compared to, for example, anti-neutrophil cytoplasmic antibody (ANCA) glomerulonephritis and vasculitis. Nevertheless, in a French study of 128 adults with IgAN, glomerular necrotizing lesions, which were seen in 9 cases, were significantly associated with development of end-stage kidney disease (ESKD) or the doubling of serum creatinine by univariate (but not multivariable) analysis [16]. In an Italian study, glomerular necrotizing lesions were seen in 35 of 340 IgAN patients and were associated with a greater probability of progression to ESKD only in patients who were not treated with immunosuppression [17]. Notably, glomerular necrotizing lesions tend to be seen more frequently in patients from Asia, perhaps related to more frequent activation of the lectin pathway of complement in these patients (see below). Shen et al. [18] examined 60 Chinese patients with active IgAN who underwent a kidney biopsy, followed by corticosteroid therapy and a follow-up biopsy within 6 months of the initial biopsy. In this cohort, 31 patients had one or more glomerular necrotizing lesions, 51 had one or more cellular or fibrocellular crescents, and 22 had E1 lesions on the initial biopsy. After steroid therapy, necrotizing lesions, active crescents, and E1 lesions were seen on 2, 15, and 5 follow-up biopsies, respectively, and none of these lesions on the initial biopsy were associated with a composite endpoint of ESKD or 30% decline in eGFR [18]. The apparent higher steroid sensitivity of necrotizing lesions compared with crescents is consistent with the former representing an earlier form of a related inflammatory process, although clearly this requires validation.

It appears that monocytes and macrophages may not play a critical role in the development of crescents in IgAN, in contrast with what occurs in other glomerular diseases, in which these cells are active players in crescent pathogenesis [19,20]. This characteristic may explain why the appearance of some or occasional crescents in kidney biopsies may not be indicative of severe lesions when compared to other forms of glomerulonephritis with crescents, unless the GBM is not preserved. In IgAN, endocapillary hypercellularity was found more frequently in cases with crescents [18], although numbers of glomerular CD68+ macrophages were found to be associated with endocapillary hypercellularity but not crescents [21]. A possible explanation for this may be the difficulty of accurate histologic diagnosis of endocapillary hypercellularity even by skillful nephropathologists, especially in the presence of segmental glomerulosclerosis that is often associated with fibrocellular crescents [21,22].

Several studies have reported that, in IgAN, complement activation plays a role in crescent formation. Complement components and factors related to complement activation are partly produced by intrinsic glomerular cells including mesangial and endothelial cells [23]. Podocytes also display complement component receptors at their cell surface. It has been reported that there is an increased intensity of properdin and factor B staining in murine IgAN with more severe glomerular injury including crescent formation, indicating the involvement of an activated alternative pathway of complements [24]. IgAN patients with increased glomerular mannose-binding lectin (MBL)-associated serine protease type 1 (MASP-1) deposition presented with higher levels of proteinuria and increased proportions of extracapillary proliferation, glomerular sclerosis, and renal dysfunction [25]. IgAN with crescents has been reported to present with higher levels of tissue C5b-9, MASP 1/3, MASP2, properdin, and factor B than IgAN without crescents, indicating an activation of both the lectin and alternative pathways of complement [26].

Glomerular C4d staining in the absence of C1q deposits is considered a typical sign of lectin pathway activation. C4d-positive IgAN biopsies are associated with a worse prognosis and a trend to develop ESKD [24]. In a cohort of 100 IgAN Chinese patients with various proportions of crescents, signs of complement activation in urine samples were significantly increased in comparison to healthy controls in cases with crescents involving >50% of glomeruli, presenting with high levels of the common complement pathway—C3a, C5a, and C5b-9—as well as markers of an alternative pathway—Bb—and the lectin pathway—C4d and MBL [27]. The levels of urinary C4d showed a highly significant linear association with the number of crescents. In a subgroup of these patients, immunohistochemistry was performed on renal biopsy tissue. Glomerular staining for C4d (>25% of glomeruli) was observed in 20% of cases in the group with crescents <25% of glomeruli, in 70% of cases with 25–49% crescents, and all cases with crescents in ≥50%. Positive glomerular staining was seen predominantly in the mesangial area, very often within the crescents and sclerotic lesions. Glomerular C5b-9 deposition and C3d staining were observed in almost all crescentic cases. These data stress the role of lectin pathway activation in Chinese patients with crescentic IgAN.

## 3. Turning from MEST to MEST-C Score

In studies of the cohort of 265 patients on which the original Oxford classification was based [28], the presence or absence of cellular/fibrocellular crescents was not a significant predictor of the rate of eGFR decline or a composite outcome of ESKD or a ≥50% decline in eGFR. This was also true in several validation studies [29,30,31,32], which like the original Oxford study excluded patients with an eGFR of <30 mL/min at the time of biopsy and/or progression to ESKD within 12 months of the biopsy. However, other studies with less restrictive entry criteria found crescents to be a prognostic indicator of a poor outcome [33,34,35,36]. Noteworthily, Katafuchi et al. found similar results to those in the original Oxford study in patients meeting entry criteria for the latter, although in their entire cohort of 702 patients and in those 286 patients not meeting the entry criteria of the original Oxford study, crescents independently predicted the development of ESKD [33].

Twenty studies published between 2009 and 2016 evaluating the association between crescents and kidney outcome involving more than 5000 patients with IgAN were included in a meta-analysis [37]. Nine of these studies [28,30,31,32,35,38,39,40,41] compared measures of kidney function between patients with no crescents or any crescents. Those patients with crescents had lower eGFR levels (*p* = 0.023); higher proteinuria (*p* = 0.024); more frequent M1 (*p* = 0.003), E1 (*p* < 0.001), S1 (*p* = 0.016), and T1/2 (*p* < 0.001) lesions; and received immunosuppressive therapy more frequently (*p* < 0.001) than those without crescents. Pooled results also showed that crescents were associated with progression to ESKD (*p* < 0.001), suggesting that potential inclusion of crescents in the Oxford Classification needed further evaluation.

In addition, two studies of pediatric patients with IgAN from Japan [35] and Sweden [36] without restrictive entry criteria also found cellular or fibrocellular crescents to be predictive of a poor outcome (eGFR <60 mL/min/1.73 m^2^ and ESKD or eGFR reduction >50%, respectively) by univariate analysis and by multivariable analysis (including eGFR, mean arterial pressure, and proteinuria at the time of biopsy) in the Japanese cohort.

In response to these findings, a working group of the International IgA Nephropathy Network (IIgANN) performed a multicenter study of the impact of crescents on renal outcomes in over 3000 patients with IgA nephropathy pooled from four previously studied, well-defined cohorts: the European VALIGA study cohort [42], two large Asian cohorts [31,33], and the original Oxford cohort [28] that included patients from four continents. Notably, while each of the three former studies validated the findings of the original Oxford study with regard to the impact of M, S, and T scores on renal outcomes, just one [33] showed crescents to be an independent predictor of a poor outcome (ESKD), although in the VALIGA study [42] an association with eGFR loss was noted only for patients not treated with immunosuppression. The working group study [10] was limited to cellular and fibrocellular (at least 10% cells) crescents, as in the original Oxford study identification of fibrous crescents showed poor inter-observer reproducibility [43]. In this study, one or more crescents were seen in 36% of biopsies (<10% crescents in 61%). The presence of crescents was associated with a faster rate of renal function decline compared to no crescents and a reduced survival from a combined event of ESKD or a ≥50% decline in eGFR in a multivariable analysis including eGFR at the time of biopsy and time-averaged proteinuria and mean arterial pressure during follow up [28]. This study also validated the association of the Oxford M, S, and T scores with the combined event in all patients although, in patients treated with corticosteroids and/or other immunosuppressive agents, only the T score remained a significant, independent predictor of increased risk of the combined event. The lack of an impact of crescents in patients treated with immunosuppressive agents is consistent with a repeat biopsy study from China that showed complete resolution of cellular/fibrocellular crescents following immunosuppressive therapy in 36/51 patients having one or more crescents on their initial biopsy [18]. Notably, however, the presence of cellular or fibrocellular crescents in ≥1/4 of the glomeruli was independently associated with a combined event even in patients who received corticosteroids and/or other immunosuppressive agents [18]. Thus, a revised version of the Oxford classification for IgA nephropathy published in 2017 includes a C (crescent) score in addition to the original MEST scores: C0 (no cellular or fibrocellular crescents), C1 (crescents in <25% of glomeruli, suggesting a poor prognosis in patients not receiving immunosuppressive therapy), and C2 (crescents in ≥25% of glomeruli) [21].

While the predictive value of crescents on renal outcomes was not evident in the initial VALIGA study [44], Coppo et al. [42] performed a follow-up study of this patient cohort based on a prolonged follow-up period (median 7 years, as compared with 4.7 years in the original study) of up to 35 years. This longer-term analysis of the VALIGA cohort, which included adults and children, showed that, in patients who never received corticosteroids/and or other immunosuppressive treatment, the presence of crescents (C1 + C2) was related to the rate of renal functional decline, independent of the MEST score and other risk factors. There was not a significant effect of crescents on the composite endpoint of 50% loss of eGFR or ESKD, perhaps because the patients enrolled had a modest median annual eGFR loss (1.8 mL/min/year) and the fraction of patients with crescents was only 10.5% (C1 = 8.6%; C2 = 1.9%).

## 4. Risk Prediction Models for IgAN and Crescents in Adults

In recent years, several models using novel mathematical statistics or artificial intelligence approaches have been developed to predict, at the time of kidney biopsy, the risk of progression of IgAN toward ESKD or 50% decline in eGFR [45]. Variables most commonly included were age, gender, blood pressure, creatinine, eGFR, proteinuria, and renal biopsy lesions according to the Oxford classification. The most recognized, due to the great number of included cases from global multiethnic cohorts, is that developed by the IIgANN, which considered two models with and without ethnicity and used clinical predictors and MEST scores at the time of renal biopsy to predict the risk of 50% decline in eGFR or ESKD at 8 years [45].

This IIgANN prediction tool was validated in external cohorts and was updated for its use also in children [46] and for its employment up to two years from the time of biopsy [47]. This tool, now available for clinical use online in a mobile-app calculator, has been recommended by 2021 KDIGO glomerulonephritis guidelines [14]. Each MEST score component was required to be entered in the prediction formula, but crescents were excluded, as their presence or absence was apparently not associated with clinical outcomes. Crescents were significantly associated with race/ethnicity (more frequent in Japanese ethnicity) and with use of corticosteroid/immunosuppressive therapy after biopsy (56% vs. 36%, *p* < 0.001). In the recently published prediction tool, at time points after renal biopsy, the value of crescents (present or absent) was re-checked, adding this variable in the post-biopsy models, but there was no improvement in in model fit or in calibration indices [47].

Other prediction models generated in different cohorts, reported the effects of crescents in untreated patients. Among 3380 Korean patients, crescents improved the discrimination performance of the prediction model only in patients not receiving corticosteroids/immunosuppressive agents [48]. However, this study did not employ the Oxford classification. A more recent report, using the MEST score, investigated 545 Korean patients to generate a prognostication model that considered C1 (found in 24% of patients) and C2 (found in 1.3%). When adding crescents to their full model with clinical data and MEST score, the predictor was not superior to the full model. However, the prediction performance of crescents was significantly improved after 5 years of follow up in the subgroup of 426 patients not treated with corticosteroid-immunosuppressive drugs (*p* < 0.02) [49].

In recent years, artificial intelligence has arisen in the medical field to assess new prediction models, claiming that machine learning techniques may outperform conventional statistical models, with an improved capability to identify variants relevant to clinical outcome. A prediction model using an artificial intelligence statistical approach developed in 2047 Chinese patients with a median of 10 years of follow up did not include crescents due to non-significance [50]. In a Caucasian cohort mostly involving the VALIGA European patients, the artificial intelligence tool showed a performance value of 0.82 in patients with a follow up of 5 years. Crescents were included, although these were not associated with significantly increased risk to develop ESKD by Cox proportional hazard models [13].

## 5. Risk Prediction Models for IgAN and Crescents in Children

The IIgANN gathered 1060 pediatric cases of IgAN from various continents. As in adults, crescents were more frequent in patients of East Asian ethnicity (65.9% of Japanese and 45.7% of Chinese versus 25.5% of Caucasians) [46]. Children received steroid therapy in 58% of the cases, more frequently in the presence of crescents (70% of children with crescents versus 48% without crescents). No association of crescents (absent versus present) with the secondary outcome of 30% reduction in eGFR or end-stage kidney disease was found, even after adjusting for immunosuppression. The prediction performance was the same in subgroups of treated versus untreated by immunosuppression. Any prediction improvement associated with crescents is confounded by the effects of other predictor variables already included in the model, such as race/ethnicity. However, the prediction model performed equally well in treated and untreated cases, indicating that the combination of risk factors contained in the models predicted outcome similarly in treated and untreated children.

In a pediatric cohort from Japan, a threshold of 30% of glomeruli with crescents was found to be predictive of development of an eGFR below 60 mL/min/1.73 m^2^ by both univariate analysis and a multivariate analysis including MEST scores and proteinuria at the time of biopsy [35].

## 6. IgAN with Crescents Involving >50% of Glomeruli

Crescentic IgAN, defined as >50% cellular crescentic glomeruli on kidney biopsy, was investigated in 113 Chinese adult patients [11]. At biopsy, the mean serum creatinine level was 4.3 mg/dL, and the mean percentage of crescents was 66%. Kidney survival rate at 5 years after biopsy was 45.8%. Multivariable Cox regression revealed initial serum creatinine as the only independent risk factor for end-stage kidney disease (*p* = 0.002). Notably, the percentage of crescents was not independently associated with ESKD.

A recent report from Japan focused on children with >50% glomeruli with crescents, accounting for 25 cases (4.9% of the whole cohort of 515 Japanese children with IgAN) [51]. No prior history of urinary abnormalities was reported in 16/25 children, who were referred from school screening programs. These children with crescentic IgAN had more frequent gross hematuria (76%), proteinuria ≥ 1 g/day, and shorter duration from clinical onset to renal biopsy, median 4 months versus 8 months. There was a significantly increased frequency of M1 (80%) and E1 (83%) in comparison with the other children, although T > 0 was rare in both groups. At the time of renal biopsy, renal function was actually well preserved (eGFR 120 mL/min) in comparison to the other children (eGFR 104 mL/min). Although eighteen children with crescentic IgA were treated with prednisolone or prednisolone plus other immunosuppressive agents, only 8/25 patients with these lesions had remission of proteinuria (6 of whom received immunosuppression) and 4 (16%) progressed to an endpoint of eGFR < 60 mL/min/1.73 m^2^ or ESKD compared to 13 (2.7%) of the remaining 490 patients, the majority of whom had C1 (n = 228) or C2 (n = 40). Children with crescentic IgAN had significantly lower survival from this composite endpoint at 13 years post-biopsy (77.1% vs. 92.6%, *p* < 0.0001). Failure of treatment to induce a remission of proteinuria and a higher percentage of tubular atrophy/interstitial fibrosis were also predictors of an unfavorable outcome [51].

## 7. Timing of Renal Biopsy and Crescents

A challenge in establishing the value of percentage of crescents as a risk factor for progression in IgAN is represented by the timing of renal biopsy, due to changes which may occur over time [52]. A study in Japanese children with IgAN [53] considered the time elapsed from the diagnosis of urinary abnormalities—mostly detected after school screening programs—and renal biopsy and reported that a shorter time from onset to renal biopsy was associated with higher glomerular percentage of crescents, in addition to higher M and E lesions. This suggests that crescents are associated with disease onset and then likely undergo a healing process into sclerotic lesions, which are commonly detected in biopsies performed years after onset.

## 8. Treatment and Crescents in IgAN

According to all the data presented in this review, the literature reports showing benefits of corticosteroid/immunosuppressive treatment in patients (adults and children) having IgAN with crescents are uncontrolled and retrospective. The coincidence of other MEST lesions with C, the variable morphology of crescents (cellular or fibrocellular, segmental or global glomerular involvement) and association with other lesions not included in the MEST scores (e.g., necrosis, interstitial inflammation) render the understanding of the role of crescents as an independent feature promoting progression of IgAN very difficult. Moreover, the reports describe patients treated with corticosteroids and other immunosuppressive drugs at varying doses and over varying intervals, with different levels of supportive care (e.g., renin–angiotensin system (RAS) inhibitors).

A recently published Argentinian–Spanish retrospective study assessed the impact of steroids plus mycophenolate in a cohort of 25 patients with progressive IgAN [54]. Progressive IgAN was defined by a decrease in eGFR of at least 10 mL/min in the 12 months prior to the start of treatment and proteinuria ≥0.75 g/day despite maximum tolerated doses of RAS blockade, and hematuria (≥5 red blood cells per high power field) at the beginning of treatment. The mean interval between the performance of kidney biopsy and the onset of therapy was 4.5 ± 11.9 months. Ten patients (47.6%) had C1 scores and three (14.3%) displayed C2 scores (patients with crescents in >50% of glomeruli were excluded). The mean duration of immunosuppression treatment was 24.7 ± 15.2 months. In the 12 months prior to treatment, the median rate of kidney function decline had been 23 mL/min/year. After the onset of treatment, the median eGFR slope was 5 mL/min/year (*p* = 0.001 with respect to the 12 months prior to treatment). Proteinuria decreased from 1.8 g/day (range 1.0–2.5) at baseline to 0.6 g/day (range 0.3–1.2) at the end of treatment (*p* = 0.01), and hematuria disappeared in 40% of patients. There were no serious adverse effects requiring treatment discontinuation. In this study, in which >50% of the population presented with active crescents in their biopsies, the addition of immunosuppression decreased the rate of decline of kidney function plus the degree of proteinuria and hematuria.

There are few data on the predictive impact of MEST-C scores in randomized clinical trial settings. An exploratory analysis was performed in 70 available renal biopsies from 162 randomized STOP-IgAN trial participants and correlated the results with clinical outcomes [55]. Kidney biopsies had been performed from 6.5 to 95 months (median 9.4) prior to randomization. This secondary analysis of STOP-IgAN biopsies indicated that M1, T1/2, and C1/2 scores were associated with worse renal outcomes. In particular, patients with glomerular crescents (C1/2 scores) in their biopsies were more likely to develop ESKD during the 3-year trial phase, but this trend was only significant in patients under supportive care.

As commented previously, consideration of crescents in managing the treatment of IgAN has been strongly discouraged by KDIGO 2021, which does not support the value of MEST score in general as a guide of treatment and, in particular, disregards the value of percentage of crescents as an independent risk factor supporting a more aggressive therapy. According to KDIGO guidelines, the presence of crescents in kidney biopsy, even when involving ≥50% of glomeruli, in the absence of a concomitant rise in serum creatinine does not constitute a rapidly progressive form of IgAN, which is instead defined as a >50% decline in eGFR over >3 months, where reversible causes are excluded. KDIGO guidelines suggest repeating renal biopsy in case of insufficient benefits from supportive care, but this is in contrast with the suggestion not to use pathology scores to guide treatment. A few uncontrolled studies have repeated renal biopsies in selected cases [18,56] after corticosteroid or other immunosuppressive drugs had been administered and showed a reduction in crescents in most treated cases. Notably, in one repeat-biopsy study, it was found that, in patients with crescents on their original biopsy, those who continued to have crescents on their repeated biopsy were significantly more likely to develop ESKD than those whose second biopsy showed resolution of crescents [56].

Recently, a single-center Chinese study investigated 140 patients enrolled between 2008 and 2016 with C1 and proteinuria <1 g/day who received supportive care (n = 52) or steroid-based immunosuppressive therapy (n = 88) [57]. The primary outcome was the rate of renal function decline. Baseline data showed a population with mild renal disease, median proteinuria of 0.6 g/day, and a median fraction of crescents of 7% (5–12%), with a follow-up time of 69 months. The rate of renal function decline was slower in the steroid-based immunosuppressive therapy group than in the supportive care group. Multivariable linear regression analyses showed that steroid-based immunosuppressive therapy significantly slowed the rate of renal function decline (*p* = 0.013) after adjusting for age, sex, mean arterial blood pressure, proteinuria, eGFR, M1, E1, S1, T1-2, the fraction of crescents, and use of RAS inhibitors. Similar findings were seen in 66 patients (33 from each group) who were matched for baseline demographic, clinical, and pathologic findings. In the matched cohort, the rate of renal function decline was also slower in the steroid-based immunosuppressive therapy group. The conclusion was that steroid-based immunosuppressive therapy may slow down the rate of renal function decline of IgAN patients with C1 and proteinuria ≤1 g/day.

Novel therapies are under evaluation for the treatment of IgAN (Table 1). These drugs are presently targeting persistent proteinuria despite several months of optimized supportive care, but the perspective is that at least some of them could be of benefit in the control of the most active and progressive cases such as those with high presence of crescents.

The agents being tested include (a) drugs blocking B cell function and survival by inhibiting B-lymphocyte stimulator (BLyS, also known as B cell activating factor, BAFF) and APRIL (a proliferation-inducing ligand); (b) complement activation inhibitors, targeting the lectin complement pathway (mannose-associated serine protease, MASP), the alternative complement pathway (factor B) or the common complement pathway (C3, C5, and its proinflammatory activation product C3a); and (c) other hemodynamic and inflammatory pathways such as endothelin, angiotensin II, nuclear factor erythroid-derived 2-related factor agonist (Table 1). Future perspectives are expected from the next results of these studies for treating patients with active histological forms of IgAN, including crescentic IgAN, independently from the persistence of proteinuria despite optimal supportive care.

## 9. Conclusions

Several factors may account for the variable results regarding the impact of crescents on clinical outcomes in IgAN. These include (a) the different patient ethnicities; (b) different timing of the renal biopsy after onset of clinical manifestations; (c) the histologic type of crescents included in each study, either cellular, fibrocellular, or fibrous; (d) the association of crescents with other histopathologic markers of activity, which blunted their independent value; (e) the varying choice of immunosuppressive treatments and dosage regimens; and (f) the different clinical outcomes.

In contrast to the case in some other forms of proliferative glomerulonephritis, the most evident conclusion from the data reported in this review is that crescents do not appear to be an independent predictor of clinical outcomes in patients with IgAN, especially those receiving corticosteroids or other immunosuppressive agents. Although the benefit of such treatment in IgAN with crescents indirectly emerges from this consideration, there is a lack of direct evidence of beneficial effect of corticosteroids or immunosuppressive drugs in every patient with IgAN presenting with crescentic lesions. Over the last decades most severely crescentic cases received corticosteroid-immunosuppressive treatment, and a placebo-controlled RCT in these patients is not conceivable. However, the value of crescents should not be disregarded. An integrated overall consideration including other MEST scores and clinical data and novel biomarkers should be undertaken in achieving a more promising personalized therapeutic approach to IgAN patients.

## Figures and Tables

**Figure 1 jcm-11-03569-f001:**
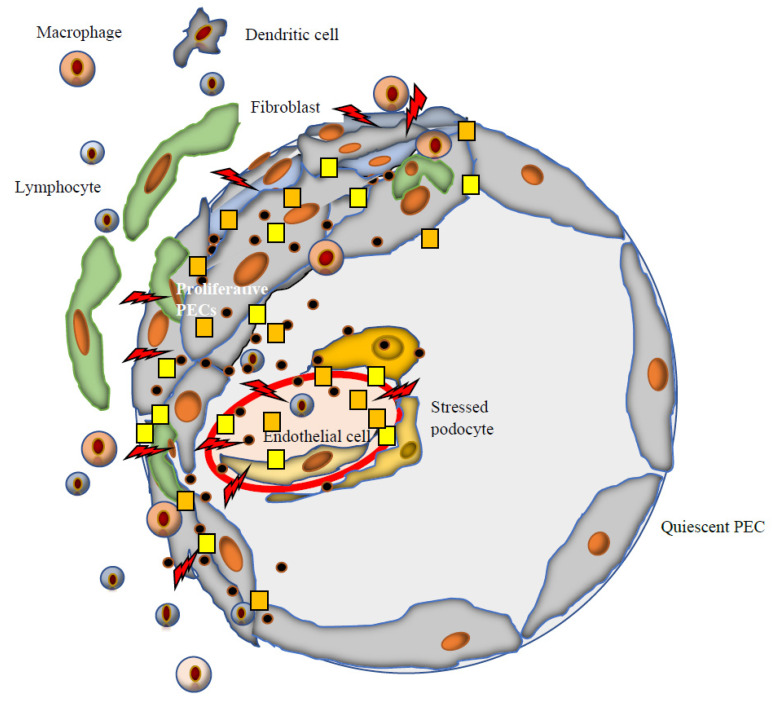
Main cells and molecules involved in crescent development in IgAN. In IgAN, crescent formation could be divided into two steps. First of all, there is an initial damage to the endothelium of the glomerular filtration barrier (rents 
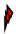
 )from where podocytes and parietal cells interact within and outside the surrounding glomerulus. The second step is the consequence of this interaction: The development of the crescent itself. Main molecules involved: Fibrinogen and fibrin-related antigens 
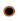
 , type IV and V collagens, laminin, fibronectin, cytokeratin persistently positive at all stages of crescents, as well as vimentin, distributed in podocytes and parietal epithelial cells (PECs). At early stages of crescent formation in IgAN, podocytes play a key role, while the accumulation of basement membrane components adds to the progression of the crescents (Figure 1). In IgAN, endothelial proliferation associates with crescent appearance. Crescent formation in IgAN is associated with activation of the lectin 
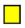
 and alternative pathways 
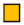
 .

**Table 1 jcm-11-03569-t001:** New drugs under evaluation for treatment of IgA nephropathy: selected from studies published in the U.S. National Library of Medicine Clinical Trials.gov.

Agent	Activity/Target	Registered Trial N (NCT)
B cell immunomodulators		
Atacicept	BLyS-APRIL inhibitor	02808429
BION-1301	APRIL inhibitor	03945318
VIS-649 Sibeprenlimab	APRIL inhibitor	04287985
RC-18	BLyS receptor inhibitor	04291782
**Complement inhibitors**		
LNP-023 Iptacopan	Factor B inhibitor	04578834
FB-LRx	Anti-sense factor B inhibitor	04014335
OMS-721 Narsoplimab	MASP inhibitor	03608033
ALN-CC5-Cemdisarin	C5 inhibitor	03841448
CCX-168 Avacopan	C5a receptor inhibitor	02384317
Ravulizumab	C5 inhibitor	04564339
APL-2	C3 inhibitor	04564339
**Various**		
CHK-01 Atrasentan	Endothelin A receptor inhibitor	04573920
Sparsentan	Endothelin and Angiotensin II receptor inhibitor	03762850
RTA-402 Bardoxolone methyl	Nuclear factor erythroid-derived 2-related factor agonist	03366337

## Data Availability

Not applicable.

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
