# Peer review of "Crescents and IgA Nephropathy: A Delicate Marriage"

_jcm, 2022, doi:10.3390/jcm11133569_

Round 1

Reviewer 1 Report

Reviewer comments

1. In response to these findings a working group of the IIgANN performed a multicenter study of the impact of crescents on renal outcomes in over 3,000 patients with IgA nephropathy pooled from four previously studied, well-defined cohorts, the European VALIGA study cohort, two large Asian cohorts and the original Oxford cohort that included patients from four continents. 
Was there any difference observed in the renal outcomes in the European VALIGA study cohort, two large Asian cohorts, and the original Oxford cohort?

2. Include a sub-section of the current clinical trials for the treatment of IgAN. List out in the form of a table. What drugs are available in the market or in clinical trials for the treatment of IgAN?

3. Include a section on future perspectives on prevention, diagnosis and the treatment of IgAN.
4. Any animal models showing the intricate relationship between crescent formation, IgAN and glomerular filtration rate (renal outcomes)?
5. Include figures for molecular factors involved in crescent formation in IgAN and tables or graphs to summarize the sections.

Minor comments:
6. Line no. 296-298: Persistent proteinuria of >2g/d and a higher percentage of tubular atrophy/interstitial fibrosis were also predictors of unfavorable outcomes. Is it >2g per day or deciliter?

Author Response

Buenos Aires, June 6th 2022

Journal of Clinical Medicine

                                On behalf of the co-authors and myself, I want thank your opportunity to submit a new version of our submitted manuscript CRESCENTS AND IgA NEPHROPATHY: A DELICATE MARRIAGE, which has been undoubtedly enriched by the reviewers’ insights. 

                                 We did our best to address all the reviewers inputs and comments, have added one more reference, plus one table and one figure, to illustrate the addressed contents.

                                    Thus, we outline the answers to the reviewers, and we hope to have fulfilled the requirements of the journal, hoping this version is found to be satisfactory.

I remain sincerely

Hernán Trimarchi

Answer to Reviewer #1.

  • In response to these findings a working group of the IIgANN performed a multicenter study of the impact of crescents on renal outcomes in over 3,000 patients with IgA nephropathy pooled from four previously studied, well-defined cohorts, the European VALIGA study cohort, two large Asian cohorts and the original Oxford cohort that included patients from four continents. 
    Was there any difference observed in the renal outcomes in the European VALIGA study cohort, two large Asian cohorts, and the original Oxford cohort?

Answer: The requested detail is now added at line 182 of first manuscript : “regard to the impact of M, S, and T scores on renal outcomes, just one [32] showed crescents to be an independent predictor of a poor outcome (ESKD), although in the VALIGA study [41] an association with eGFR loss was noted only for patients not treated with immunosuppression”

2) Include a sub-section of the current clinical trials for the treatment of IgAN. List out in the form of a table. What drugs are available in the market or in clinical trials for the treatment of IgAN?

Answer: as requested in this point and in the following point 3 we added the perspective of treatment with new drugs in form of a table with comments (Table I and Line 374 first manuscript).

“Novel therapies are under evaluation for the treatment of IgAN (Table I). These drugs  are presently targeting persistent proteinuria despite several months of optimized supportive care, but  the perspective is that at least some of them could be of benefit in the control of the most active and progressive cases as those with high presence of crescents.

The agents being tested include a) drugs blocking B cell function and survival by inhibiting B-lymphocyte stimulator (blys), B cell activating factor (BAFF) and APRIL (a proliferation inducing ligand); b) Complement activation inhibitors, targeting the lectin complement pathway (Mannose Associated Serine Protease, MASP), the alternative complement pathway (Factor B) or the common complement pathway (C3, C5 and its proinflammatory activation product C3a); c) other hemodynamic and inflammatory pathways such as endothelin, angiotensin II, nuclear factor erythroid-derived 2 related factor agonist (Table I). Future perspectives are expected from the next results of these studies for treating patients with active histological forms of IgAN, including crescentic IgAN, independently from the persistence of proteinuria despite optimal supportive care.”

  • Include a section on future perspectives on prevention, diagnosis and the treatment of IgAN.

Answer:  Treatment options are included in the answer to point 2. The area of prevention and diagnosis of IgAN seems too broad to be treated in this Review focused on crescentic IgAN.

4) Any animal models showing the intricate relationship between crescent formation, IgAN and glomerular filtration rate (renal outcomes)?

Answer.  Unfortunately, the experimental models of IgAN are only partially like human IgAN mostly due to differences in mice of the IgA molecular structure. Moreover, almost all experimental models of IgAN do not show crescents, but mostly mesangial proliferation and glomerular sclerosis. We added this specification, after your request in the text, line 77 of the first manuscript and the relevant recent publication (new ref 15).  Monteiro RC, Suzuki Y. Are there animal models of IgA nephropathy? Semin Immunopathol. 2021 Oct;43(5):639-648.

  1. Include figures for molecular factors involved in crescent formation in IgAN and tables or graphs to summarize the sections.

Answer Figure 1 is now provided as you suggested

  1. Line no. 296-298: Persistent proteinuria of >2g/d and a higher percentage of tubular atrophy/interstitial fibrosis were also predictors of unfavorable outcomes. Is it >2g per day or deciliter?

Answer:: the value was >2 g/day We amended  the sentence line 296 first manuscript  in a simpler  and more readable format.

Reviewer 2 Report

Many thanks for this comprehensive overview of the evidence base informing crescentic IgA Nephropathy. I just have a few minor suggestions for consideration:

Lines 118 - 121: is it possible to comment on this further? E1 is associated with monocyte/macrophages and crescents are not - a postulation for why then an association between the two exist would be interesting to read.

Line 175-176: would be helpful to know what was corrected for on multiivariate analysis here.

Line 178: IIgAN is defined further down the manuscript but appears here first, it would be better defined here. 

Line 272: would be helpful to know what was included in the multivariate analysis here 

299 - 307: could the brevity between clinical presentation and biopsy be explained by more severe disease creating an urgency for biopsy, and thus more crescents? From what I can see this is not easily deducible in the reference, but may account for the finding

Line 356-357: Could this be because of the time lag alluded to in lines 299-307, and not due to a true treatment effect? Maybe worth commenting on for the benefit of the reader given the studies were uncontrolled. 

Lastly, both TESTING and STOP-IgAN included a cohort of IgAN patients with crescents, it maybe worth discussing the outcomes of this cohort too. 

Many thanks again.

Author Response

Buenos Aires, June 6th 2022

Journal of Clinical Medicine

                                On behalf of the co-authors and myself, I want thank your opportunity to submit a new version of our submitted manuscript CRESCENTS AND IgA NEPHROPATHY: A DELICATE MARRIAGE, which has been undoubtedly enriched by the reviewers’ insights. 

                                 We did our best to address all the reviewers inputs and comments, have added one more reference, plus one table and one figure, to illustrate the addressed contents.

                                    Thus, we outline the answers to the reviewers, and we hope to have fulfilled the requirements of the journal, hoping this version is found to be satisfactory.

I remain sincerely

Hernán Trimarchi

Answers to Reviewer n 2

  • Lines 118 - 121: is it possible to comment on this further? E1 is associated with monocyte/macrophages and crescents are not - a postulation for why then an association between the two exist would be interesting to read.

Answer We further commented this point adding an interesting recent report at line 118 first manuscript.In IgAN, endocapillary hypercellularity was found more frequently in cases with crescents [17], although numbers of glomerular CD68+ macrophages were found to be associated with endocapillary hypercellularity but not crescents [20].  A possible explanation for this may be the difficulty of accurate histologic diagnosis of endocapillary hyperpecellularity even by skillful nephropathologists, especially in the presence of segmental glomerulosclerosis that is often associated with fibrocellular crescents [20,21]. 

  • Line 175-176: would be helpful to know what was corrected for on multiivariate analysis here.

Answer: as you suggested we added the specification about the factors used to correct the multivariate analysis in the Japanese cohort:multivariable analysis (including eGFR,  mean arterial pressure, and proteinuria at the time of biopsy) in the Japanese cohort”.

  • Line 178: IIgAN is defined further down the manuscript but appears here first, it would be better defined here. 

Answer: Indeed the acronym should be specified, and, as you suggested we added it: International IgA Nephropathy Network (IIgANN)

  • Line 272: would be helpful to know what was included in the multivariate analysis here 

Answer: Indeed the sentence was not sufficiently clear. We simplified it as follows:
     In a paediatric cohort from Japan, a threshold of 30% of glomeruli with crescents was found to be predictive of development of an eGFR below 60 ml/min/1.73 m2 by both univariate analysis and a multivariate analysis including MEST scores and proteinuria at the time of biopsy [34].

  • 299 - 307: could the brevity between clinical presentation and biopsy be explained by more severe disease creating an urgency for biopsy, and thus more crescents? From what I can see this is not easily deducible in the reference, but may account for the finding

Answer: Your observation is acute and careful, but children were mostly detected by school screening programs and their clinical features were mild. Hence the Authors did not perform a search for correlation between clinical onset severity of time elapsed to renal biopsy. We added the specification “….the diagnosis of urinary abnormalities - mostly detected after school screening programs

  • Line 356-357: Could this be because of the time lag alluded to in lines 299-307, and not due to a true treatment effect? Maybe worth commenting on for the benefit of the reader given the studies were uncontrolled.

Answer. Following your comment we added a specification:Notably, in one repeat biopsy study it was found that in patients with crescents on their original biopsy, those who continued to have crescents on their repeat biopsy were significantly more likely to develop ESKD than those whose second biopsy showed resolution of crescents [56]. 

  • Lastly, both TESTING and STOP-IgAN included a cohort of IgAN patients with crescents, it may be worth discussing the outcomes of this cohort too. 

Answer: Our Review selectively focuses on crescentic IgAN   Both TESTING and STOP-IgAN trials were not aimed at investigating the value of crescents in response of corticosteroid /IS therapy. The difference in ethnicity, baseline MESTC score, time elapsed before renal biopsy, rate of GFR decline and treatment modalities are too broad to engage a comparison of outcomes in the present review article. We are afraid that expanding the text with this difficult analysis might make things even less clear than what we tried to present.

Reviewer 3 Report

The authors report an interesting and complicated topic regarding the role of crescent in IgA nephropaty in relation to management and outcome.

The review is well written and I suggest adding a table with the major studies that have treat IgAN with immunosoppressive therapy

Author Response

Buenos Aires, June 6th 2022

Journal of Clinical Medicine

                                On behalf of the co-authors and myself, I want thank your opportunity to submit a new version of our submitted manuscript CRESCENTS AND IgA NEPHROPATHY: A DELICATE MARRIAGE, which has been undoubtedly enriched by the reviewers’ insights. 

                                 We did our best to address all the reviewers inputs and comments, have added one more reference, plus one table and one figure, to illustrate the addressed contents.

                                    Thus, we outline the answers to the reviewers, and we hope to have fulfilled the requirements of the journal, hoping this version is found to be satisfactory.

I remain sincerely                                                                                                                                                                                                                                                 Hernán Trimarchi

Answers to Reviewer n 3

The review is well written and I suggest adding a table with the major studies that have treated IgAN with immunosuppressive therapy

Answer: Thank you for your favorable comment. As you may tell, the various therapies related to our topic, crescents in IgAN, have been discussed throughout the paper. Moreover, we have added Table 1, in which all then novel therapies under clinical assessments have been outlined in Table 1, that hope will fulfill your petition.

Round 2

Reviewer 1 Report

The authors have addressed all the comments to improvise the manuscript.